# Cardiovascular Magnetic Resonance Imaging in Familial Dilated Cardiomyopathy

**DOI:** 10.3390/medicina59030439

**Published:** 2023-02-23

**Authors:** Clement Lau, Uzma Gul, Boyang Liu, Gabriella Captur, Sandeep S. Hothi

**Affiliations:** 1New Cross Hospital, Royal Wolverhampton NHS Trust, Wolverhampton WV10 0QP, UK; 2Institute of Cardiovascular Sciences, College of Medical and Dental Sciences, University of Birmingham, Birmingham B15 2TT, UK; 3MRC Unit for Lifelong Health and Ageing at UCL, University College London, London WC1E 6BT, UK; 4Institute of Cardiovascular Science, University College London, London WC1E 6BT, UK; 5Centre for Inherited Heart Muscle Conditions, Cardiology Department, The Royal Free Hospital, London NW3 2QG, UK

**Keywords:** familial dilated cardiomyopathy, cardiovascular magnetic resonance imaging, DCM, CMR

## Abstract

Dilated cardiomyopathy (DCM) is a common cause of non-ischaemic heart failure, conferring high morbidity and mortality, including sudden cardiac death due to systolic dysfunction or arrhythmic sudden death. Within the DCM cohort exists a group of patients with familial disease. In this article we review the pathophysiology and cardiac imaging findings of familial DCM, with specific attention to known disease subtypes. The role of advanced cardiac imaging cardiovascular magnetic resonance is still accumulating, and there remains much to be elucidated. We discuss its potential clinical roles as currently known, with respect to diagnostic utility and risk stratification. Advances in such risk stratification may help target pharmacological and device therapies to those at highest risk.

## 1. Introduction

Cardiomyopathies are myocardial disorders in which the heart is structurally and functionally abnormal. They are currently sub-classified on the basis of cardiac morphology as hypertrophic cardiomyopathy (HCM), dilated cardiomyopathy (DCM), arrhythmogenic cardiomyopathy (ACM), and left ventricular noncompaction cardiomyopathy (LVNC). The European Society of Cardiology (ESC) divides dilated cardiomyopathy into two groups, familial and nonfamilial [1]. Conventionally, idiopathic DCM with at least one first- or second-degree relative with confirmed idiopathic DCM is defined as familial DCM [1,2]. These non-ischaemic cardiomyopathies are less common than ischaemic cardiomyopathy [3]. Of non-ischaemic aetiologies, idiopathic DCM is the second most common aetiology, accounting for 31% of cases. Non-ischaemic DCM is more common in female, non-white, and younger individuals [3].

Dilated cardiomyopathy accounts for up to one-third of heart failure cases and is one of the leading causes for cardiac transplantation. The reported prevalence of DCM from epidemiological data is 36.5/100,000 individuals in Western populations [3]. This is likely to be an underestimate, however, since its diagnosis has relied on modalities with low sensitivity, such as echocardiography or angiography. Hershberger and colleagues, in their review, were able to report a higher prevalence of DCM of > 1 per 250 individuals on the basis of recent data [1,4].

Estimations of the prevalence of *familial* DCM range widely, between 2% to 65%, averaging 23% in a meta-analysis of 23 studies [5]. This is due partly to heterogeneity of the diagnostic criteria as well as increasing diagnosis over time related to more systematic clinical screening [5]. Familial DCM has the following subtypes with already mapped genetic loci (>40): autosomal dominant, autosomal recessive, X-linked, and mitochondrial forms. These might comprise either a pure cardiomyopathy or may have associated myopathy [6,7]. The penetration of familial DCMs is incomplete, variable, and age-dependent [6,7].

Amongst familial DCM, monogenic causes account only for approximately 30–40% of cases [8,9]. The implication of this is that that traditional Mendelian considerations will leave more than half of cases without a defined monogenic cause. It is, therefore, likely that complex mechanisms underlie familiar DCM rather than monogenic transmission alone. It has accordingly been proposed that common variants predispose to DCM in the appropriate environmental exposure, while rarer variants may underlie monogenic forms. Including rare variants in the genetic panel increases the yield of genetic testing for DCM, with a genetic diagnosis achieved in approximately 40% of apparently familial cases [4]. Pathogenic genetic variants can be identified in 15–25% of sporadic DCM [8,9]. Titin (*TTN*) mutations are the most common aetiology of familial DCM, occurring in ~25% of familial cases of DCM and in 18% of sporadic cases [1].

Rare variants in more than 30 genes can produce a DCM phenotype, some of which also underlie other cardiomyopathies, inherited muscle diseases, or myopathic syndromes. These genes encode both contractile and non-contractile proteins, such as cytoskeletal proteins, abnormalities of which result in the DCM phenotype. This results in reduced resistance to mechanical stress as well as abnormalities of intracellular calcium handling, myocellular energetics, and sarcolemmal ion channel function [1]. De novo mutations are rare and defined when none of the biological parents carry the offspring’s mutation and confirm the pathogenic status of genetic variants [9]. A multicentre study tested the hypothesis that both familial and non-familial DCM have a rare variant genetic basis and concluded that most idiopathic DCM have a genetic basis [10].

Specific genetic mutations predict the unique course of DCM. Presentations with arrhythmias and premature conduction disease preceding DCM are often associated with *Lamin A/C* (*LMNA*) mutations [4]. DCM associated with sarcomere mutation is characterised by earlier disease onset and prominent ventricular arrhythmias, while *SCN5A* and desmin gene (*DES*) mutations present with conduction disease or ventricular arrhythmias as the dominant features [4].

## 2. Diagnosis

The diagnostic process consists of a detailed clinical history alongside at least a three-generation family pedigree, comprehensive cardiac imaging, biochemical profile, and genetic testing, where clinically indicated. Systematic screening with electrocardiography (ECG) and echocardiography of first-degree relatives of patients with idiopathic DCM has been proposed to identify subclinical forms. European Society of Cardiology Guidelines recommend screening with an ECG and echocardiogram in all first-degree relatives of an index patient with DCM, irrespective of family history [11].

Mestroni et al. [6] define *familial* DCM by either the presence of two or more affected relatives within a single family or in the presence of a first-degree relatives of a DCM patient with SCD below age 35 years [6]. Criteria proposed to diagnose the index case includes imaging criteria, family history, ECG, imaging criteria, and exclusion of competing causes, such as significant coronary artery disease, chronic alcohol excess, uncontrolled hypertension, persistent arrhythmia, pericardial disease, congenital heart disease, and cor pulmonale. There are some inherent difficulties in identifying the index case. Firstly, the varying clinical presentation and course of the disease lends diagnostic challenges. Secondly, acquired disorders, such as hypertension, excess alcohol intake, and systemic inflammatory diseases may produce phenocopies of idiopathic DCM or act as environmental factors, unmasking rare variants [6].

## 3. Genetics in Familial DCMs

Studying the genetic basis of DCM requires either multigeneration DCM pedigrees or genome-wide sequencing. Most studies use the latter approach, and once DCM-associated variants are identified, the numbers of DCM patients with these variants are compared with the number of individuals in the reference datasets carrying the same variants. A probability of 90% or more is required to be labelled pathogenetic, although this does not necessarily reflect causation [1]. Sequencing of large numbers of genes is required due to low prevalence, heterogeneity of mutations, private mutations, modifier genes, and different mutations producing the same phenotype [4]. Fortunately, there are large, easily available genetic datasets which enable evaluation of already identified pathogenic mutations. Another approach, used by most clinical centres, is to perform targeted next generation sequencing of high-risk variants.

Several genome-wide sequencing association studies (GWAS) have identified genetic variants associated with DCM. However, the yield has been limited by modest sample sizes (< 5000 cases). The prevalence of these pathogenic genetic variants is greater than the estimated disease prevalence. Hershberger suggested this mismatch is due either to lower penetrance of the mutations, non-pathogenicity of some, or that the actual DCM prevalence is higher than estimated [4]. Villard et al. [12] were the first to perform genome-wide study, while Esslinger et al. [13] and Meder et al. [14] carried out similar studies involving 3000 and 4000 DCM cases with matched controls, respectively.

## 4. Cardiac Imaging in Familial DCMs

DCM has been defined by echocardiography by the degree of systolic impairment, that is, fractional shortening (FS) less than 25% (> 2SD) and/or ejection fraction less than 45% (> 2SD) and the degree of LV enlargement; or LV end diastolic diameter (LVEDD) greater than 117% (2SD (112%) plus 5%) or end diastolic volume (LVEDV) greater than 2SD of the predicted value, as corrected for age and body surface area, excluding any known cause resulting in the myocardial abnormality observed [6].

Echocardiography is often the first imaging test for assessing LV remodelling and also provides associated data, such as the presence and severity of functional mitral regurgitation. Speckle-tracking echocardiography uses the distinct speckle pattern in the myocardium to assess myocardial deformation. Abnormalities of strain and strain rate can be detected by echocardiography in first-degree relatives of patients with DCM, indicating a subclinical phenotype [15].

Cardiovascular magnetic resonance imaging (CMR) is the reference standard for measurement of ventricular volume, ejection fraction, and myocardial mass. In addition, CMR detects myocardial oedema, which, when present, may suggest an inflammatory basis for the observed phenotype. Long native myocardial T1 time and high extracellular volume (ECV) fraction may be helpful in differentiating DCM from athletic heart adaptation or iron overload cardiomyopathy. The presence, pattern, and burden of late gadolinium enhancement (LGE) may be helpful in determining the risk of malignant ventricular arrhythmias. Echocardiography may be suboptimal in certain individuals when CMR is recommended as an alternative. Progressive increases in chamber dimensions, strain abnormalities, and LGE are features of early DCM [16]. Longitudinal studies over many years with imaging are required to characterise DCM progression in genetically predisposed individuals. CMR offers higher repeatability in volumes and ejection fraction and may allow detection of subtle changes in surveillance of mutation carriers. LGE detects replacement fibrosis but not diffuse fibrosis, so it may be deceptively reassuring even in those with established diffuse fibrosis, where native T1 is long and ECV is high.

Amin et al. [2] demonstrated that the combining CMR with genetic information allows better DCM stratification and results in a change in management, as per an ESC Position Paper [2]. In DCM, the degree of fibrosis shown by LGE, is a predictor of mortality and hospitalisation, particularly ventricular arrhythmias [17]. Hyper-trabeculation is also detected commonly in DCM (36%), although its presence was not associated with adverse outcomes, and this can be detected in normal hearts [18,19].

Imaging traits of DCM have been used to investigate genetic variants involved in a DCM phenotype [18,20]. valuation of approximately six thousand subjects from cardiac imaging from the Candidate Gene Association Resource (CARe) Study and more than a thousand individuals from the Multi-Ethnic Study of Atherosclerosis (MESA) with CMR yielded four associated genes [21]. Pirrucello et al. [7] analysed CMR-derived left ventricular measurements in 36,000 UK Biobank and 2000 Multi-Ethnic Study of Atherosclerosis participants, identified 45 new loci, and developed a polygenic score for prediction of DCM on the basis of variants most strongly associated with DCM phenotypic variables on CMR.

## 5. CMR Sequences for DCM Assessment

The cardiac MRI protocol for DCM involves cine imaging and tissue characterisation, including mapping and gadolinium-enhanced images; flow mapping, oedema, and stress perfusion imaging may be considered. Cine images use a balanced steady-state free processing (SSFP) sequence, which provides high signal-to-noise ratio and myocardial blood pool contrast or flash (spoiled gradient echo) sequence in case of cardiac device or at 3T. These are usually retrospectively ECG gated along with breath-held-in expiration, with a minimum of 25 reconstructed phases, with a slice thickness and gap of 8 mm and 2 mm to give temporal resolution less than or equal to 45 ms and spatial resolution 2 mm.

Gadolinium-enhanced late images are inversion recovery or phase-sensitive inversion recovery gradient echo sequences, segmented as a routine, while single shot for patients with difficulty holding their breath or irregular rhythm. In plane resolution, it should be ~1.4–1.8 mm. These are generally acquired 10 min after the contrast injection; however, the delay should be tailored to the dose and protocol; images are acquired every other beat but can vary according to heart rate. A TI scout can be used as a guide for choosing inversion time to null myocardium; however, inversion time can be different due to different readout parameters. The late gadolinium-enhanced images are acquired in the same planes and slices as cine images.

T1 mapping sequences most frequently involves shortened MOLLI (ShMOLLI) during breath-holding, with the trigger delay set to end-diastole, slice thickness 6–8 mm, with in-plane resolution 1.6–2 mm. The native and 15 min post contrast maps are acquired as short axis slice, but number and plane of slices can be modified. Other sequences, such as SASHA, which involve saturation instead of inversion, are also available.

T2 mapping uses either a T2-prepared SSFP sequence or fast spin echo sequences, usually with motion correction.

Black blood T2-weighted short tau inversion recovery (STIR) sequence is used for oedema imaging. Images are usually acquired in three short axis slices as well as long axis views.

## 6. CMR Characteristics of Familial DCMs

Septal linear mid-wall fibrosis is characteristic of, although not specific to, familial DCM. Furthermore, its presence carries adverse prognosis [22]. In a meta-analysis by Becker et al., LGE was present in approximately 45% of patients with DCM and was associated with adverse outcomes, including ventricular arrhythmia, hospitalisation, and death, while its absence predicted LV reverse remodelling [20]. Table 1 summarises the features of genetic DCM subtypes, which are now detailed further by subtype.

### 6.1. Lamin A/C Cardiomyopathy

*LMNA* mutations can manifest as isolated LV dilatation, isolated LV dysfunction, or a typical DCM phenotype, but marked dilatation and wall thinning is not characteristic and does not recur post-transplantation [23].

A meta-analysis of 299 lamin A/C mutation carriers showed a high risk of ventricular arrhythmias, heart failure, and a SCD rate of up to 46% [22]. In a multicentre study of 269 *LMNA* mutation carriers, male patients appear to have a higher prevalence of ventricular arrhythmias and progression to end-stage heart failure [24]. There is a higher risk of tachyarrhythmias and both sinoatrial node and atrioventricular node conduction disease at young age, as well as arterial and venous thromboembolism [23]. The disease may progress from atrial arrhythmias to conduction disease to a hypokinetic, nondilated cardiomyopathy to DCM. Diastolic dysfunction, reduced septal longitudinal strain, and increased mechanical dispersion of strain by speckle tracking are present, irrespective of ejection fraction, and they predict the development of conduction abnormalities and arrhythmias [25]. Biventricular involvement can occur. Prominent right ventricular epicardial fat may be seen on CMR, and on LGE images, basal to mid septal, mid-myocardial fibrosis is a common (Figure 1A) and early finding that correlates with diastolic dysfunction [26]. Mid-wall LGE in the basal to mid septum is a common and early finding in lamin A/C cardiomyopathy, which is associated with conduction disease and ventricular arrhythmias [27]. Moreover, myocardial fibrosis by CMR appears to correlate with PR interval prolongation and predicting ventricular arrhythmias [28].

Extracellular volume is increased early, while extensive fibrosis is not found until late in the disease course [27,29]. With *LMNA* mutations, regional wall motion abnormalities are typical in the basal segments and correlate with the presence and degree of LGE [27]. ECG features of lamin heart disease include various mild electrophysiological phenotypes, such as first-degree atrioventricular block, fragmented QRS complexes or tall R waves in the anterior or septal chest leads, complete or incomplete bundle branch block, or intraventricular conduction delay. Screening of first-degree relatives with ECG and echocardiography (with Holter monitoring if conduction disease is present in the proband) should commence starting at 10–12 years of age.

### 6.2. Duchenne and Becker Muscular Dystrophies

Duchenne muscular dystrophy (DMD) and Becker muscular dystrophy (BMD) are X-linked disorders affecting the synthesis of dystrophin. Dystrophin has an important role in stabilizing the cell membrane and transmits forces generated by sarcomere contraction to the extracellular matrix. Dystrophin-associated cardiomyopathy features a mild reduction in left ventricular systolic function, without morphologic abnormalities in early childhood. Disease progression leads to the typical phenotype of a dilated LV with global hypokinesia. Regional wall motion abnormality and thinning of the basal to mid lateral segments, less often the inferior wall or septum, are also seen. There may be mid-wall or subepicardial fibrosis in the same segments, with transmural fibrosis in advanced disease [30]. The subepicardial fibrosis of the inferolateral wall is often seen despite preserved global LV systolic function in early DMD, while the advanced stage has extensive subepicardial lateral wall as well as septal fibrosis. BMD manifests with severe and earlier cardiac disease than DMD [30,31]. The severe childhood form manifests with an increased right precordial ECG R-to-S ratio, deep Q waves in the lateral leads, conduction abnormalities, and mainly supraventricular, but also ventricular, arrhythmias [31].

The pattern of LGE in BMD is similar to DMD, with initial involvement of the subepicardial inferolateral wall, followed by age-dependent progression of both fibrosis and systolic dysfunction. Thus, transmural late enhancement of the lateral wall and mid-myocardial septal fibrosis are consistent with advanced disease [31]. LV remodelling index, the ratio of end diastolic mass to volume, differentiates between dilated and hypertrophic cardiomyopathy, with the latter being the common phenotype in Duchenne muscular dystrophy [30]. LVEF remains the most powerful predictor of adverse outcomes. Soslow and co-workers showed shorter postcontrast T1 ratio and increased myocardial native T1 and extracellular volume detected in the absence of LGE and despite normal LVEF in a small number of DMD patients [32]. Starc et al. [33] reported significantly increased global ECV in 47 DMD patients (29 ± 6%) compared with published normal values (24 ± 2%). Wansapura et al. [34] reported, in their small sample size of 26 DMD cases, that T2 heterogeneity quantified by FWHM increased progressively over time as the systolic function declined. Gaur et al. [35] found muscle–fat fractions were significantly higher in skeletal muscle, but not in myocardium; this warrants further investigation into fat–water composition in DMD affected myocardium. Thus, native T1 values may be low in regions of fatty metaplasia. Reduced circumferential strain has been found in patients with normal LV size and function [36].

Disease-specific studies in patients with muscular dystrophy also highlight the prognostic power of LGE. In 88 male patients with BMD and DMD, transmural LGE combined with an LVEF ≤ 45% independently predicted hospitalization for heart failure and/or occurrence of non or sustained ventricular tachycardia after a mean follow-up time of 47 ± 18 months [37]. In a small retrospective study of 32 patients with Duchenne muscular dystrophy, higher LV end-systolic volume and lower LVEF were associated with increased mortality [38]. LGE by CMR was also an independent predictor of adverse cardiac remodelling, ventricular arrhythmias, and mortality.

Since myocardial fibrosis detected by LGE imaging (Figure 2, and Appendix A) may be observed when echocardiography is normal, and early introduction of heart failure therapies may lead to reverse remodelling, CMR should be considered as part of screening for genotype-positive individuals. Cardiac screening is recommended for female mutation carriers, starting after the teenage years, given the risk for developing cardiomyopathy [31]. CMR may be included in the screening regimen since it may reveal myocardial fibrosis in female mutation carriers [31].

Other Dystrophin mutation diseases include Emery–Dreifuss muscular dystrophy (EDMD), caused by mutations in the nuclear membrane proteins. The myocytes are replaced by fibrosis and fat starting in the atria (thus early atrial arrhythmias), then eventually affects the ventricles (progressive dilatation and systolic failure) and atrioventricular node (conduction abnormalities and sudden death presentation) [31]. Cardiac screening, including female carriers, is recommended. Cardiac MRI does not show fibrosis in early-stage disease, although subtle abnormalities are seen through measurement of reduced systolic circumferential strain [30,31]. Cardiac involvement occurs in limb–girdle muscular dystrophy with a DCM phenotype, and mid-basal septal LGE can be seen before systolic dysfunction and LV dilatation manifest [31]. X-linked DCM phenotype of dystrophin and related mutations has preferential cardiac involvement without any overt skeletal muscle involvement, and the disease course may be rapidly progressive, according to some reports [39].

### 6.3. Arrhythmogenic Cardiomyopathy

Mutations in filamin-C (*FLNC*) are responsible for both arrhythmogenic cardiomyopathy (ACM) and DCM, and they are characterized by non-ischaemic fibrosis, ventricular arrhythmias, and risk of sudden death [40]. The fibrosis detected by LGE is reported in the subepicardial (or mid-myocardial) inferolateral wall, with abnormal regional wall motion with or without focal wall thinning. Electrocardiography may show low voltages and flat or inverted T waves in the inferolateral or lateral leads [40,41].

In a cohort of 89 patients with DCM, Augusto et al. showed that *DSP* and *FLNC* genotypes appears to be associated with subepicardial, ring-like LGE pattern not seen in other DCM and, thus, representing a specific ACM phenotype [41]. Moreover, LV impairment in *DSP*/*FLNC* is related to scarring and is more regional, whilst non-DSP/FLNC genotypes have overall less scarring and greater impairment of LVEF and GLS. This highlights the potential role that CMR has in the initial identification of high-risk genotypes.

There are several *SCN5A* mutations identified, which are associated with a DCM-arrhythmia and overlap syndromes. For example, the c.665G>A gain-of-function sodium channel mutation results in a phenotype of multifocal premature contractions and DCM [42]. The R222Q mutation with resultant decrease in sodium current is also identified in familial DCM. With the c.74A > G missense mutation, focal intramyocardial LGE in the inferoseptal wall has been reported in a family cluster; however, no specific MRI features are reported for other mutations [43].

### 6.4. Sarcomeric Cardiomyopathies

Sarcomere gene mutations are associated with other cardiomyopathies in addition to DCM, with decreased calcium sensitivity and contractility leading to a DCM phenotype. Most common mutations associated with autosomal dominant DCM are in β-myosin heavy chain and troponin T. Autosomal recessive DCM is caused by mutations in cardiac troponin I, while cluster of families with Troponin C mutation-related DCM have been reported [44]. Clinically, DCM caused by sarcomere mutations is indistinguishable from other forms of idiopathic DCM since it lacks any red flags apart from a disease course dictated by age of presentation [44]. It can present from infancy to adulthood. Presentation early in life is associated with adverse outcomes, including sudden cardiac death and refractory heart failure [45]. On the other hand, adult-onset sarcomere DCM usually has a milder course and slow or no progression [41,45]. DCM associated with *TNNC1* and *TNNT2* mutations is 100% penetrant and is often severe [41,45]. In subclinical mutation carriers, subtle abnormalities in systolic function have been reported [41,45]. Linear mid-myocardial fibrosis on LGE may or may not be present, but increased T1 values represent underlying diffuse fibrosis [44].

Desmin (*DES*) mutation is an uncommon cause of DCM, with a frequency of DES gene mutation of < 1% reported in FDCM [44,46]. It has autosomal dominant inheritance and can present with atrioventricular conduction abnormality or ventricular arrhythmias in addition to heart failure. Although the most common phenotype is reported to be restrictive cardiomyopathy, both DCM and arrhythmogenic cardiomyopathy (ACM) phenotypes (with biventricular involvement) also occur [47]. CMR in early disease shows focal left ventricular hypertrophy not detected by echocardiography and focal fibrosis detected by LGE imaging [47,48]. Both subepicardial and mid-myocardial late gadolinium enhancement has been reported in the lateral, anterior, and inferoseptal LV walls, apex, and right ventricle [49,50]. Advanced cases show regional wall motion and extensive transmural scarring, and the extent of late enhancement is related to poor prognosis. *DES* p.Q113_L115del has been associated with DCM, with markedly increased left ventricular trabeculation [48]. There are reports of oedema detected by T2 images; thus, active inflammation may occur. Perfusion sequences may demonstrate ischemia [50]. Presentation can occur with both skeletal and cardiac muscle involvement or an isolated cardiac phenotype [50]. Skeletal involvement occurs with progressive distal limb and respiratory muscle weakness, and it may precede or follow the cardiac presentation [47,48]. CMR is superior to echocardiography for the detection of the early, subclinical stage in desmin mutation carriers [47]. The site of mutation defines the time of cardiac presentation and severity; for example, R406W is associated with severe and early cardiac involvement [49].

Titin mutations are most commonly associated with autosomal dominant inheritance. Cardiac involvement occurs in 50%, with congenital heart disease, dilated cardiomyopathy phenotype, or arrhythmias [51]. Extra cardiac involvement includes skeletal and respiratory muscle involvement and syndromic facies. Cardiac involvement (Figure 1D) is more likely in mutations impacting both N2BA and N2B, especially truncating mutations [51]. Cardiac screening is strongly recommended for all titin manifest patients and should be considered in carrier relatives, particularly with truncating mutations [51].

Tayal and colleagues showed that CMR-derived LVEF, presence of mid-wall LGE, and indexed left atrial volume determines prognosis in *TTN* DCM, with no significant differences compared with other aetiologies of DCM [52,53]. Mutations in the *TTN* have been shown to predispose patients to malignant ventricular arrhythmias. In 319 DCM patients undergoing CMR, 13% were *TTN*-positive [54]. Moreover, compared with *TTN*-negative DCM patients, those who were *TTN*-positive had significantly lower LVEF, thinner LV walls, and higher incidence of sustained VT, independent of LVEF [54]. A similar-sized study of 303 DCM patients also found 13% of the cohort carrying the *TTN* mutation [55]. The authors showed that, compared with *TTN*-negative patients, the *TTN*-positive counterparts had lower indexed LV mass by CMR despite similar cardiac size and function and had increased interstitial fibrosis on endomyocardial biopsy, with an increased risk of ventricular arrhythmias at median follow up of 45 months [55]. Despite this, another study of 83 patients with *TTN*-positive DCM did not find a significant difference in medium term prognosis (~4 years) compared with DCM patients without *TTN* [52].

### 6.5. Mitochondrial Disease

Most mitochondrial diseases are caused by mutations in the nuclear DNA (nDNA), with only 15% caused by mutations in the mitochondrial DNA (mtDNA), which are transmitted through the maternal line [53,56]. Whereas MELAS presents mainly with hypertrophic cardiomyopathy, MERRF and Kearns Sayre can also present with DCM (although less common), along with conduction disease or Wolf Parkinson White syndrome [53,56]. Atrioventricular block is very prevalent in Kearns Sayre disease. Extra cardiac involvement, including neurological, ophthalmological, and endocrine, are common [53]. Diffuse fibrosis may be seen earlier; however, extensive LGE is indicative of end-stage disease and poor response to treatment [53,56]. There is a paucity of large studies with long-term follow up to determine the prognostic value of LGE detected in these patients.

### 6.6. Glycogen Storage Disorders

Danon disease may be associated with DCM less commonly than HCM phenotype. Late gadolinium enhancement has been reported in the LV, sometimes extensive, with sparing of mid-septum, but with involvement of right ventricular insertion points [57]. Increased values for native T1 and ECV have been reported [57].

## 7. Risk Stratification by CMR

### 7.1. Late Gadolinium Enhancement

Traditionally, stratifying patients with a LVEF ≤ 35% as having an increased risk of morbidity and mortality have not translated to benefits in non-ischaemic DCM patients receiving implantable cardiac defibrillators [58]. Myocardial tissue characterisation by CMR has proven to be a powerful tool in predicting the risk of ventricular arrhythmias and sudden cardiac death (SCD) in patients with familial DCM. Risk stratification with LGE appears to hold prognostic power. The location and pattern of LGE appears to be associated with increased risk of SCD in DCM. Halliday et al. followed 874 patients over a median of 4.9 years and found that septal LGE, even in small amounts, is associated with a large increase in the risk of death and SCD [59]. The risk of SCD is greatest with both septal and free-wall LGE present. Several large cohort studies have shown that the presence of LGE independently predicts an increased risk of hospitalisation, SCD, and all-cause mortality [60,61,62]. In a large meta-analysis of 4554 DCM patients, those with LGE compared with those without LGE had increased cardiovascular mortality (odds ratio (OR) 3.40; 95% CI: 2.04 to 5.67), ventricular arrhythmias (OR: 4.52; 95% CI: 3.41 to 5.99), and rehospitalisation for HF (OR: 2.66; 95% CI: 1.67 to 4.24) [20]. These findings were corroborated by a prospective study of 1165 patients followed up for a median of 3 years, in whom LGE was a strong predictor of ventricular arrhythmias or sudden death. Furthermore, increased risk was associated with epicardial, transmural, and combined septal and free-wall LGE. Interestingly, patients with LVEF > 35% and LGE had the greatest risk (annual event rate 3%; *p* = 0.007) [63].

The absence of LGE in patients with DCM increases the likelihood of left ventricular recovery (LV reverse remodelling). In a meta-analysis of 4554 patients, the absence of LGE was a strong independent predictor of LV reverse remodelling (OR, 0.15; 95% CI: 0.06 to 0.36) at 2 years [64]. The extent of LGE at follow-up was associated with progressive LV dysfunction [64].

### 7.2. T1 Mapping and Extracellular Volume Assessment

Diffuse fibrosis by native T1 mapping may hold prognostic value. In a prospective longitudinal study of 131 cardiomyopathy patients (59 non-ischaemic), native T1 mapping was an independent predictor of ventricular arrhythmia [65]. Later, a larger study of 637 consecutive patients with DCM, with a median follow-up of 22 months, native T1, and ECV, the presence and extent of LGE were predictive of all-cause mortality and heart failure (all *p* < 0.001) [66]. Moreover, ECV may have even greater prognostic power than native T1 mapping. Vita et al. [67] followed 240 non-ischaemic cardiomyopathy patients after a median of 3.8 years. The authors found that for every 10% increase, mean ECV portended to a 2.8-fold adjusted increased risk to major adverse cardiovascular events (*p* < 0.001) [67]. Furthermore, ECV in the anteroseptum appeared to have the greatest risk [67].

### 7.3. Strain Imaging

Longitudinal strain assessed by feature-tracking CMR analysis may also have utility in predicting risk in DCM patients. In a study of 210 patients with DCM followed up for a median of 5.3 years, GLS as measured by CMR had independent and incremental prognostic value, surpassing traditional biomarkers, such as NT-proBNP, LVEF, and LGE. GLS of > −12.5% predicted outcome, even in patients with LVEF < 35% (*p* < 0.01) and in those with LGE (*p* < 0.001) [68].

### 7.4. Right Ventricular Characteristics

Right ventricular structure and function is complex and difficult to accurately assess. CMR is currently the imaging modality of choice to assess RV function with high reproducibility. Adverse RV remodelling is common in HF and DCM, and studies are continuing to unravel the clinical relevance of such changes [69,70]. In 250 DCM patients followed up over a median of 6.8 years, RV systolic dysfunction (RVSD), defined by a RV EF ≤ 45% by CMR, was a significant independent predictor of all-cause mortality or cardiac transplantation in DCM patients (hazard ratio, 3.90; 95% CI: 2.16–7.04; *p* < 0.001) [71]

## 8. Role of CMR in Guiding Device Therapy

In patients with DCM, the presence of LGE in CMR coupled with an LVEF ≤ 50% suggests consideration of an implantable cardiac defibrillator (ICD) for primary prevention [72]. In a study of 452 patients with non-ischaemic cardiomyopathy and LVEF < 35% on optimal medical therapy who met the criteria for ICD insertion, ICD reduced all-cause mortality (HR, 0.45; 95% CI: 0.26–0.77) and cardiovascular death (HR, 0.51; 95% CI: 0.27–0.97) when LGE was present after a median follow-up period of 37.9 months [73]. The DANISH-MRI study, a pre-specified sub-study of the DANISH trial (152), recruited 252 patients with non-ischemic DCM and an indication for primary-prevention ICD on optimal medical management [74]. Overall prognosis in the patients with LV scarring was worse, but ICD implantation did not significantly reduce all-cause mortality in those with LV scarring compared with those without LV scarring (HR, 1.18; 95% CI: 0.59–2.38 vs. HR 1.00; 95% CI: 0.39–2.53, *p* for interaction = 0.79) [74].

CMR can also be used to predict and guide response to device therapy. Chalil et al. [75] derived the tissue synchronization index (CMR-TSI) determined by segmental radial wall motion data, generating tissue synchronization polar maps of the LV, which is a global dyssynchrony measure. A CMR-TSI ≥ 110 ms was an independent predictor of death or unplanned hospitalisation in a cohort of 77 HF patients referred for cardiac resynchronisation therapy (CRT) after a median follow-up of 2 years [75]. The presence of LGE may also guide lead placement in CRT. In 60 patients receiving CRT implantation, RV and LV leads over scarring was associated with non-response to CRT [76]. LV lead positions at areas of LGE compared with viable myocardium were associated with an increased risk of CV death (HR 6.34; *p* < 0.0001) or HF hospitalizations (HR 5.57; *p* < 0.0001) [77].

## 9. Future Directions

Diffusion tensor imaging by CMR assesses the motility of the sheetlets that form the microarchitecture of the LV. In a small study of 19 DCM patients, compared with 13 healthy controls, sheetlet function was abnormally characterised by altered systolic conformation and reduced mobility [78]. Khalique et al. [79] then showed that these changes might persist in patients with DCM with recovered LV function measured by a reduction in LV size and improvement in LVEF [79]. Thus, diffusion tensor imaging may identify patients at risk of relapse after remission, although larger studies are required.

Flow assessed by 4D-CMR to ascertain the forces on the myocardium through LV filling were investigated in a small study of 10 stable DCM patients versus 10 healthy controls [80]. The authors found that the ratio of forces were heterogenous, acting in the long and short axis directions and magnitude during LV diastole in DCM patients. Furter studies using 4D-flow CMR may provide insights into abnormal LV filling and subclinical phenotypes of DCM.

## 10. Conclusions

More than half of familial DCM cases have an undefined genotype, even though rapid advances in genetic analysis have improved the diagnostic yield. Genotype determines the unique course and features of DCM, and, thus, genetic diagnosis may significantly influence management. Identifying the genetic basis of DCM may differentiate phenocopies of DCM, improve family screening pathways, provide insight into disease pathogenesis, and trigger investment into novel treatment strategies directed at molecular targets. CMR accurately identifies the DCM phenotype, reliably detects the early subtle changes in cardiac structure and function in the silent stage, and provides characteristic pointers towards an underlying genotype. CMR has a fundamental position in the diagnosis and management of familial DCM along with genetic testing. CMR provides disease-specific imaging biomarkers for risk stratification, including strain, ECV scarring detected by LGE. This may reflect a tendency for reverse remodelling as well as adverse events in DCM, such as arrhythmias, SCD, and hospitalisation due to heart failure. Novel CMR techniques continue to provide insight into the underlying pathophysiology of cardiomyopathy and perhaps detect subsets of patients with a tendency to recurrent drop in systolic function after recovery. Finally, CMR may identify patients to be prioritised for genetic analysis and help target specific DCM cohorts that might benefit most from a defibrillator or resynchronisation device, creating a more individualised DCM management approach.

## Figures and Tables

**Figure 1 medicina-59-00439-f001:**
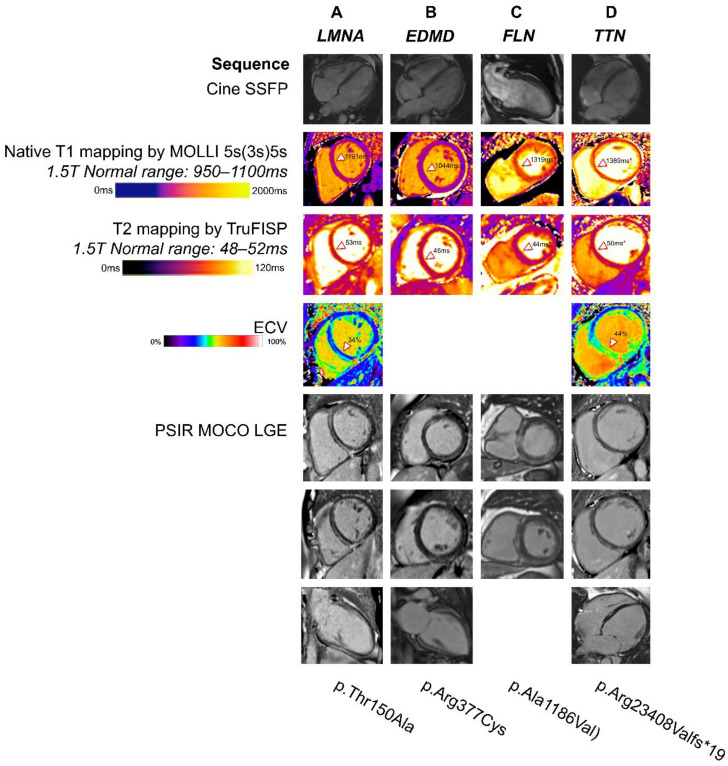
CMR multiparametric characteristics in non-ischaemic DCM. (**A**) Male patient with LMNA-related DCM showing regional prolongation of native T1 time in the septum (interstitial expansion) along with mild oedema and diffuse fibrosis. Typical pattern of mid-wall fibrosis in the septum. (**B**) Female patient with EDMD and cardiomyopathy with CMR showing normal biventricular size with mildly impaired LV systolic function and basal septal and inferior mid-wall fibrosis. (**C**) Male patient with FLNC mutation, positive family history of SCD, and CMR showing moderately impaired LV function and prominent ring of subepicardial fibrosis from basal to mid LV. (**D**) Male patient with TTN mutation and CMR showing dilated, globular, and dyscoordinate LV with moderate systolic impairment and basal septal mid-myocardial LGE and basal lateral epicardial LGE. * Scans performed at 3 T where different normal ranges apply.

**Figure 2 medicina-59-00439-f002:**
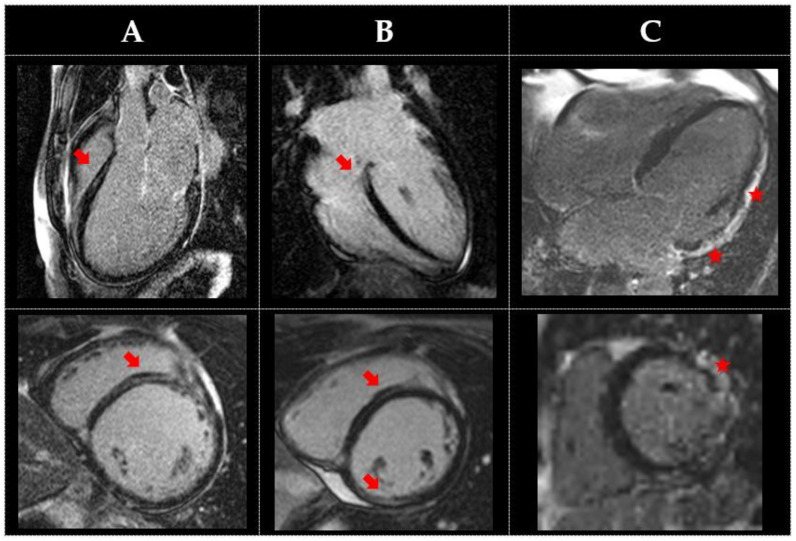
Examples of differing patterns of late gadolinium enhancement in genetic DCM, with typical pattern of mid-wall (arrows) and subepicardial LGE (star). (**A**) A 35 year-old male patient with a likely pathogenic variant in TTN (heterozygous for c.78493C > T nucleotide substitution leading to premature termination of translation in titin). The left ventricle is severely dilated with biventricular systolic impairment. (**B**) A 28 year-old female patient with a likely pathogenic variant in LMNA (heterozygous for c.513 + 1G > A nucleotide substitution, is predicted to lead to aberrant splicing). There is normal LV size and function but increased LV trabeculation, which does not fulfil Petersen criteria for non-compaction. (**C**) A 50 year-old male patient with a likely pathogenic variant of DSG2 (heterozygous for c.136C > T nucleotide substitution, leading to defective protein cleavage). There is biventricular dilatation and severe systolic impairment.

**Table 1 medicina-59-00439-t001:** Genetic dilated cardiomyopathies and their clinical, MRI, and arrhythmia features.

Mutation	Screening and Transmission	Age of Presentation/Prognosis	ECG and Clinical Features	Extra-Cardiac Features	Imaging Features/Volume and Function	Tissue Characterization	Arrhythmia/Conduction Disease Risk
Lamin	Recommended commencing 10–12 years by ECG and Echo Autosomal dominant	Young age onset and progression	Conduction or depolarization abnormalities, such as IVCD/BBB/ Early abnormalities in strain or diastolic function	Arterial and venous thromboembolism	Isolated LV dilation or dysfunction; marked dilation unusual Wall thinning not present RWMAs (basal) in Lamin A/C HNDCM to DCM progress	Prominent right ventricular epicardial fat Early increase in ECV LGE shows basal to mid septal, mid-myocardial fibrosis (extensive fibrosis only late in disease)	Increased arrhythmias and both AV and SA nodal conduction disease
Dystrophin (DMD and BMD)	Cardiac screening recommended in mutation carrier females along with affected males Consider CMR in screening alongside ECG and echo X linked	Childhood onset, progression from HNDCM to typical DCM. BMD has an earlier and more severe disease course than DMD	Increased right precordial ECG R-to-S ratio Deep Q waves in the lateral leads Early abnormalities in strain	Typical skeletal muscle weakness and disability in affected May be none in carriers	Wall thinning and RWMAs lateral wall, inferior, or septum Early increased ECV, while still normal function DCM or HCM phenotype	Mid-wall or subepicardial fibrosis in hypokinetic segments (inferolateral) in early disease (sometimes with preserved LV function) Fibrosis becomes extensive or transmural in advanced stages	Conduction abnormalities and tachyarrhythmias (both SVT and VT) Early onset in severe forms
Dystrophin (EDMD and others)	Screening recommended		Early strain abnormalities	Typical skeletal muscle weakness and disability	DCM phenotype but no specific features	Fibrosis not early feature in EDMD Septal mid-wall fibrosis early feature in limb–girdle type	Tachyarrhythmias and conduction abnormalities
Sodium channel							Arrhythmias
Filamin C	Cardiac screening recommended in mutation-positive or first-degree relatives of proband Consider CMR Autosomal		Low voltage and flat or inverted T waves inferolaterally Early LGE with normal echo ECG features of ACM absent in ACM phenotype		DCM or ACM (LV) phenotype	Subepicardial or mid-myocardial inferolateral fibrosis “Ring-like” circumferential fibrosis, but not in all cases	Ventricular arrhythmias with a more marked and malignant course than average DCM cases
Sarcomere	Screening recommended early age Exact gene determines inheritance mode and penetration	Severe and progressive disease if childhood onset Mild and non-progressive if adult onset	Indistinguishable from others since no red flag features		DCM or HCM phenotype	Mid-wall, linear fibrosis may or may not be present.	
Desmin	Screening recommended Autosomal dominant CMR may be considered alongside ECG and Echo	Variable onset and prognosis depending on exact mutation		May or may not have skeletal muscle involvement	RCM, ACM, or DCM phenotype Focal wall hypertrophy early disease Oedema in acute phase Perfusion defect	Focal fibrosis by LGE (mid-wall or subepicardial) anywhere in LV (apex, anterior, septal, lateral, or inferior) or RV Transmural scar in advanced stage disease	AV conduction abnormalities and ventricular arrhythmias
Danon	X-linked dominant			Syndromic facial features	HCM or DCM phenotype	LGE scar can be extensive and spares the mid-septum. High ECV/T1	
Titin	Cardiac screening recommended affected and carriers Autosomal Dominant			Skeletal muscle, syndrome features	CHD or DCM Thinner walls, lower LVEF		Arrhythmias
Mitochondrial	Autosomal recessive, maternally inherited or dominant			Neurological, endocrine, and ophthalmic	HCM or DCM phenotype depending on type of disease	Diffuse fibrosis (T1/ECV) Extensive fibrosis only I end stage	Conduction abnormalities and Wolf Parkinson White syndrome AVB prevalent in Kearns Sayre

## Data Availability

Not applicable.

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
