# Peer review of "Cardiovascular Magnetic Resonance Imaging in Familial Dilated Cardiomyopathy"

_medicina, 2023, doi:10.3390/medicina59030439_

Round 1

Reviewer 1 Report

In the current review, the authors summarized the importance of CMR among patients with DCMP. The paper was well written in general. The specific findings were defined in detail. I only have minor comment regarding the implementation of demonstrative pictures of CMR in each specific finding appropriately.

Author Response

Thank you for your positive comment and review. We have included two new figures demonstrative specific mutation related imaging features in the revised manuscript and widened the authorship in the process to achieve this.

Reviewer 2 Report

This paper is useful in discussing the known diagnostic utility of echocardiography and cardiac magnetic resonance imaging and the potential role of clinical risk stratification. However, the title only mentions cardiac magnetic resonance imaging and does not mention echocardiography, so this article logically needs to be further modified.

Author Response

Thank you for your kind comments. WE have modified the manuscript to reduce the discussion of echocardiography, and keep the review focused on CMR, except where comments about echo are of central importance.

Reviewer 3 Report

The authors reviewed about familial DCM (and other genetic cardiomyopathy) from the point of CMR findings. The manuscript was well-written and well-structured about the topics about CMR finding in DCM. There are some comments below to empower the readers' understandings.

1.      The authors addressed general background and its diagnosis in the first three paragraphs (“Introduction”, “Diagnosis”, “Genetics in familial DCMs”). These parts might be redundant, I would like to recommend to summary these basic backgrounds and to add general explanation of CMR methods/sequences which mentioned later parts in the manuscript.  

2.      The paragraphs “CMR characteristics of familial DCMs” and “Genotype-phenotype association and prognostic risk” might be repetitive.

3.      To readers’ comprehension, it might be better to add the table including phenotype/genotype (mutations)/CMR findings/clinical implications.

4.      If available, please add the figures of representative CMR images.

Author Response

The authors reviewed about familial DCM (and other genetic cardiomyopathy) from the point of CMR findings. The manuscript was well-written and well-structured about the topics about CMR finding in DCM. There are some comments below to empower the readers' understandings.

Thank you for the many constructive comments which we have acted upon in the revised manuscript.

  1. The authors addressed general background and its diagnosis in the first three paragraphs (“Introduction”, “Diagnosis”, “Genetics in familial DCMs”). These parts might be redundant, I would like to recommend to summary these basic backgrounds and to add general explanation of CMR methods/sequences which mentioned later parts in the manuscript.

Thank you. We have shortened these sections accordingly and removed redundancy and repletion. We have also added a section on CMR methodology/sequences.

  1. The paragraphs “CMR characteristics of familial DCMs” and “Genotype-phenotype association and prognostic risk” might be repetitive.

We have accordingly removed the repetition.

  1. To readers’ comprehension, it might be better to add the table including phenotype/genotype (mutations)/CMR findings/clinical implications.

We agree and have add a table to this effect.

  1. If available, please add the figures of representative CMR images.

We agree and have added two representative image panels.

Round 2

Reviewer 3 Report

The authors have revised adequately the manuscript according to my comments. The table and graphical figures have been very informative and comprehensive for readers. I have no additional comment.